# Nutritional Composition of Some Commonly Available Aquatic Edible Insects of Assam, India

**DOI:** 10.3390/insects13110976

**Published:** 2022-10-24

**Authors:** Mintu Sarmah, Badal Bhattacharyya, Sudhansu Bhagawati, Kritideepan Sarmah

**Affiliations:** Department of Entomology, Assam Agricultural University, Jorhat 785013, India

**Keywords:** entomophagy, nutritive value, edible aquatic insects, nutritional profiling

## Abstract

**Simple Summary:**

Many people believe that edible insects could be a good source of protein. Entomophagy is mostly practiced in India’s northeastern states. This region is home to a large number of ethnic groups or tribes with extensive traditional knowledge of edible and therapeutic insects. In addition to the nutritional benefits provided by edible insects, the functional properties and potential applications as texturizing food ingredients and ingredients of protein-rich meat replacement products must be investigated. The current findings indicate that the selected aquatic insect species are ideal candidates for further investigation as a food and feed alternative.

**Abstract:**

The nutritive value of five edible aquatic insects of Assam—Hemipterans; water bug (*Diplonychus rusticus* Fabricius) family belostomatidae; giant water bug (*Lethocerus indicus* Lepeletier and Serville) family belostomatidae; water scorpion (*Laccotrephes* sp.) family nepidae, water stick (*Ranatra* sp.) family nepidae; Coleopterans diving beetle (*Cybister* sp.) family dytiscidae—based on their proximate and elemental composition, antioxidant and antinutritional properties were assessed by using standard methods of analysis. Analytical studies revealed that the selected aquatic insect species have high nutritive value and are rich sources of protein (50.03 to 57.67%) and other nutrients (fat, carbohydrate and crude fiber, etc.) along with superior energy contents (331.98 to 506.38 kJ/100 g). The aquatic insect species also contained appreciable amounts of major and trace dietary elements. Phenol and flavonoid contents reflect its high antioxidant activity (80.82 to 91.47% DPPH inhibition). Tannin (18.50 to 60.76 mg tannic acid equivalent/100 g), phytic acid (11.72 to 97.30 mg/100 g) and oxalic acid (2.93 to 5.34 mg/100 g) as antinutritional compounds were registered below the toxic level (0.52% or 520 mg/100 g). The present findings indicate that the selected aquatic insect species can be considered as ideal candidates for exploration as food and feed to ensure nutritional and livelihood security of this region.

## 1. Introduction

Many of the world’s indigenous populations rely heavily on edible insects as a food source [1]. With an increase in global population comes an increase in consumer demand for protein, making sustainable meat production a serious challenge for the future [2]. In this regard, insects are one of the best alternative protein sources with a low environmental impact [3]. Entomophagy is primarily practiced in developing countries. Entomophagy has become ingrained in the culture of many South Asian countries, where 40% of the population is chronically malnourished [4]. Insects are consumed by nearly 2 billion people worldwide and are an important part of traditional diets [3].

Coleoptera account for 31% of all insect orders consumed globally, followed by Lepidoptera (18%) and Hymenoptera (14%). Orthopteran, Hemipteran and Isopteran insects are also popular as edibles in various parts of the world [3]. Insects are a rich and diverse source of high and healthy nutrients, containing high levels of fat, protein, vitamin, fiber and mineral content. When it comes to the amino acid composition of edible insects, phenylalanine and tyrosine are two of the most nutritionally valuable amino acids. Some insects contain a significant amount of lysine, tryptophan, and threonine, which are all lacking in certain cereal proteins [5]. The protein content of edible insects is in the range of 35–60 g/100 g dry weight or 10–25 g/100 g fresh weight [6,7], which is higher than the protein content of cereal, soybeans, and lentils [8]. Insects contain more protein than meat or chicken eggs in the upper range [9]. Orthoptera edible insects (crickets, grasshoppers, and locusts) have a high protein content [10]. However, because insects have a hard exoskeleton, their protein digestibility varies greatly [11]. Chitin-rich exoskeletons are particularly difficult to digest [12]. At the moment, we do not know whether humans can digest chitin [13]. Of course, as part of the processing, the exoskeleton can be removed [14]. According to some studies, without the exoskeleton, insect protein digestibility ranges between 77% and 98% [15]. The nutritional value of edible insects varies greatly even within the same species group, and it also depends on metamorphic stage, habitat, diet, and sex [10,16]. Aside from having high nutritional values, edible insects can also be considered a feasible and practicable venture and means of generating income, particularly in rural areas, where edible insects are cultivated and reared as mini-livestock for human food and animal feed [17].

In India, entomophagy is practiced mostly in the North-eastern states because a large number of ethnic tribes who possess a vast traditional knowledge and wisdom on edible and therapeutic insects are inhabitants in this region. Being a biodiversity hotspot, North-East India also harbors a wide array of both terrestrial and aquatic insects that can further be explored for edible purposes. Aquatic insects become more abundant in various water bodies of this region during the rainy season.

Aquatic insects, such as giant water bugs and water beetles, are collected from the wild and sold in local markets, especially in the Indian states of Manipur, Nagaland, and Assam. Similarly, aquatic insects, such as giant water bugs, water striders, and backswimmers, are popular delicacies in many Southeast Asian countries [12,18]. Even nymph and adult mayflies, which are reported to contain more crude protein [19], are widely accepted in many parts of China, Japan, New Guinea, and Vietnam.

Aquatic insects are increasingly being used as water quality indicators because they respond to a wide range of environmental conditions [20]. While we are learning more about terrestrial insect entomophagy, the same cannot be said for aquatic insects [17]. However, it is possible to cultivate aquatic edible species using the most basic of materials, such as creating shallow artificial ponds to attract migrating adult water-beetles or raising odonates, or placing flat tiles (or pieces of ‘astro-turf’) in rivers and streams to colonize by net-spinning caddisflies or blackflies [21].

Though entomophagy is practiced throughout India’s north-eastern region, there is a scarcity of scientifically validated information on the nutritional composition of edible aquatic insect fauna. A review of the available literature reveals that research on this topic is still in its early stages, so there is very little information available about the nutritional quality of edible insects and even less about edible aquatic insects. As a result, the current study was undertaken to evaluate the nutritive value of some selected species of edible aquatic insects based on their proximate composition, elemental composition, antioxidant properties, and antinutritional properties using standard methods of analysis, with the goal of establishing aquatic insects as a potential alternate source of nutrition.

## 2. Materials and Methods

### 2.1. Sample Collection

The present investigation was designed to evaluate proximate composition, elemental composition, antioxidant properties and antinutritional properties of five different aquatic insect species, viz., *Diplonychus rusticus*, *Cybister* sp., *Lethocerus indicus*, *Laccotrephes* sp. and *Ranatra* sp., which were collected using kick net method and nylon net pond method from streams, ponds, rice fields and shallow water bodies from Majuli river island (26°45′ N to 27°12′ N latitude and 93°39′ E to 94°35′ E longitudes) and Jorhat (26°44′ N latitude and 94°10′ E longitude) district of Assam. The collected aquatic insects were later killed by freezing at 0 °C, and the sorted species were washed and cleaned of dust before being sun-dried and then oven-dried at 103 °C for 4 h. The insects were then refrigerated in airtight containers with proper labels until further testing. To keep the edible portion, the dried aquatic insects’ wings and legs were removed with scissors. After that, the edible portions of each insect species were pulverized separately in a mechanical grinder to obtain a fine powder. The insect powder was sieved with a fine pore strainer to obtain a homogeneous powder, which was then stored in airtight plastic containers for further biochemical analysis.

### 2.2. Biochemical and Elemental Analysis

The five aquatic insect species were biochemically analyzed using standard procedures. Moisture content, crude fat and crude fiber were analyzed for proximate analysis using the method described in the Association of Official Analytical Chemists methods, 2000 [22] and were expressed as percentages (g/100 g). The moisture content of a fresh insect sample was determined after drying it in a drying oven (Universal Hot Air Oven, Ambala Cantt, India) for 4 h at 103 °C. Crude fat, crude fiber, carbohydrate and crude protein, on the other hand, were determined on a dry weight basis (d.w.). The carbohydrate content of the samples was estimated using the Anthrone Method, as described by Hedge et al. [23], in which the sample is treated with 80% ethanol to remove sugars and then the saccharides is extracted with perchloric acid (HClO_4_). Saccharides are hydrolyzed to glucose and dehydrated to hydroxymethyl furfural in a hot acid medium. When combined with anthrone, this compound produces a green product. Crude protein was calculated by multiplying the measured nitrogen content by a factor of 6.25 and expressed as a percentage (g/100 g) using the Micro-Kjeldahl method [24]. The AOAC method, 2000 [22] was used to determine crude fat using the Soxhlet method with petroleum ether (40 °C to 60 °C). For crude fiber determination the sample was boiled with 1.25% dilute sulfuric acid (H_2_SO_4_) and then washed with water. It was further boiled with 1.25% dilute sodium hydroxide (NaOH) and the remaining residue after digestion was taken as crude fiber. The ash content was determined using the AOAC method of 1970 [25] and expressed as a percentage (g/100 g). The energy content was calculated by multiplying the percentage of crude protein, crude fat and carbohydrate by the calorific value, i.e., by the factors 4, 9 and 4, respectively, and the results were recorded as KJ/g [26]. The mineral solution was made using the method described in the AOAC method, 1970 [25]. Atomic absorption spectroscopy was used to determine the mineral elements sodium (Na), potassium (K), copper (Cu), zinc (Zn), manganese (Mn), iron (Fe), calcium (Ca), magnesium (Mg) and sulfur (S) (AAS) Model No. iCE 3500 AA Spectrometer (Wide PMT) Atomic Absorption Spectrophotometer (True Double Beam Optics) [27]. The Vanadomolybdate method [28] was used to determine the phosphorus (P) content. The mineral composition of the insect species was expressed in mg/100 g. Using a standard protocol, the antioxidant and antinutritional properties of five aquatic insect species were determined. The antioxidant activity was determined using the 2,2-Diphenyl-1-picrylhydazyl (DPPH) method and was expressed as % DPPH inhibition, the total phenolics content was determined using Malick and Singh [29] and was expressed as mg catechol equivalent/100 g, and the total flavonoid content was determined using Woisky and Salatino [30] and was expressed as mg quercetin equivalent/100 g, with minor modifications. The tannin contents were determined using Folin–Denis reagent as described by Makkar et al. [31] and, in this method, a standard calibration curve was prepared and the Absorbance (A) against concentration of tannins at 725 nm were expressed as mg tannic acid equivalent/100 g, the phytic acid content by Wheeler and Ferrel [32] and oxalic acid was determined by Dye [33] and were expressed as mg/100 g.

### 2.3. Data Analysis

The statistical analysis for biochemical parameters was carried out using a completely randomized block design with three replications. The data were statistically analyzed, and the means were compared using Duncan’s Multiple Range Test (DMRT) [Statistical Package for the Social Sciences (SPSS) v 20].

## 3. Results

### 3.1. Proximate Composition

The estimation of the proximate composition of the five aquatic insect species is presented in Table 1. Among the five aquatic insect species, *D. rusticus* showed the highest crude protein (57.67 g/100 g) and the highest ash content (4.74 g/100 g); and *Cybister* sp. showed the highest carbohydrate (3.68 g/100 g), the highest crude fat (28.95 g/100 g) and the highest energy (2118.69 kJ/100 g). *L. indicus* showed the lowest moisture content, the lowest crude protein content and the lowest ash content, but showed an intermediate amount of carbohydrate content, crude fat and energy content. *Laccotrephes* sp. showed the highest moisture content (9.19 g/100 g), whereas its carbohydrate content and energy content were found to be the lowest. *Ranatra* sp. showed an intermediate amount of moisture, carbohydrate, crude protein, ash and energy content, whereas it showed the lowest crude fat content.

### 3.2. Elemental Composition

Macroelements such as sodium (Na), phosphorus (P), potassium (K), calcium (Ca), magnesium (Mg) and sulfur (S) content were significantly different in all the five insect species (Table 2). *D. rusticus* showed the highest content of Na (28.62 mg/100 g) and Mg (45.20 mg/100 g). *Cybister* sp. showed the highest P (153.32 mg/100 g) and K (34.60 mg/100 g), and the lowest Ca and S among the five insect species. *L. indicus* showed an intermediate macroelement content. In the case of *Laccotrephes* sp., Ca (56.15 mg/100 g) and S contents (26.45 mg/100 g) were found to be the highest among the five species, whereas there was a significant difference in P content. *Ranatra* sp. showed a significant difference in Na, K and Mg contents among all species.

Microelements such as iron (Fe), zinc (Zn), manganese (Mn) and copper (Cu) also varied among the insect species (Table 3). *D. rusticus* showed the highest Zn (7.22 mg/100 g) and Mn (4.22 mg/100 g) contents among all species. *Cybister* sp. showed the lowest Fe and Zn contents. *Lethocerus indicus* registered the lowest Mn and Cu contents, whereas *Ranatra* sp. showed the highest Fe (112.10 mg/100 g) and Cu contents (4.20 mg/100 g). However, *Laccotrephes* sp. showed an intermediate microelement content.

### 3.3. Antioxidant Properties

In the present investigation, *D. rusticus* and *L. indicus* showed intermediate total phenolics, flavonoid and percent DPPH inhibition. *Cybister* sp. showed the highest total phenolics (363.80 mg catechol equivalent/100 g) and flavonoid (50.82 mg quercetin equivalent/100 g) contents among the five insect species. *Laccotrephes* sp. showed the lowest total phenolics and % DPPH inhibition. However, *Ranatra* sp. showed the highest DPPH inhibition (91.47%) and the lowest flavonoid content among the five insect species (Table 4).

### 3.4. Antinutritional Properties

As evident from the experimental results, *D. rusticus* showed intermediate tannin, phytic acid and oxalic acid contents. *Cybister* sp. showed the lowest phytic acid content among the five insect species. *L. indicus* registered the lowest tannin and the highest oxalic acid contents (5.34 mg/100 g), while *Laccotrephes* sp. showed the highest phytic acid content (97.30 mg/100 g). Among the five insect species *Ranatra* sp. showed the highest tannin (60.76 mg tannic acid equivalent/100 g) and the lowest oxalic acid content (Table 5).

## 4. Discussion

According to the current study, the moisture content of all five aquatic insect species is less than 10%, which reduces the risk of microbial contamination and thus increases the preservation period. This is consistent with previous studies of moisture content in edible insects from Zambia [34] and Assam, India [35]. In contrast to the current study, a higher moisture content was found in aquatic insect species (13.46 to 49.05 g/100 g, f.w.) from Manipur, India [36], as well as in a few other terrestrial edible insect species (60 to 71 g/100 g, f.w.) from the Netherlands [37], which could be due to environmental factors, such as the insects’ geographical location and habitat. The carbohydrate content was 2.74 to 3.68 g/100 g, which was consistent with the findings of Shantibala et al. [36], who revealed a carbohydrate content in five aquatic insect species from Manipur, India, ranging from 0.06 to 2.39 g/100 g, d.w. Moreover, similar findings have been reported by other researchers working on various edible insects [35,38,39]. Proteins perform a variety of functions in the body, including tissue building, as well as acting as enzymes and hormones [40]. Most edible insects have a higher ratio of crude protein content on a mass basis than other animals and plant foods, according to nutritional analysis. The current study backs up previous findings that crude protein content ranged from 30.25 to 84.56 g/100 g, d.w. in twenty Assam edible insect species [35], 22.64 to 70.48 g/100 g, d.w. in five Manipur aquatic insect species [36], and 68.54 to 79.33 g/100 g, d.w. in five Assam soil-dwelling scarabs [41]. Several previous researchers observed a more or less similar trend in crude protein content in various edible insect species [34,38,42,43,44]. Because the daily protein requirement for an adult is 0.66 g/kg body weight [45], and the edible insects in this study have a high crude protein content, it can be assumed that they can be used as a supplement to meet the human protein requirement. The current study found crude fat content ranging from 8.67 to 28.95 g/100 g, which is consistent with previous findings [34,35,38,46]. The higher fat contents found in *Cybister* sp. and *D. rusticus* (28.95 and 27.87 g/100 g, d.w., respectively) may contribute to human and animal energy requirements. Lower crude fat contents (4.00 to 5.50 g/100 g, d.w.) were found in five soil-dwelling scarabs [41]. The crude fiber content of the five insect species ranged from 2.48 to 12.68 g/100 g, with the variation possibly due to chitin in the exoskeleton. The crude fiber content found in this study was consistent with previous findings [36], in which most edible insect species had a crude fiber content within the desirable range. Despite the presence of the enzyme chitinase in human gastric juices, chitin is an indigestible fiber [47]. However, it was discovered that this enzyme was inactive. People from tropical countries with a long history of eating insects have a higher level of active chitinase in their bodies [48]. Chitin removal improves insect protein digestibility [49]. In the current study, the ash content on a dry weight basis ranged from 2.39 to 4.74 g/100 g. The lower ash content of the insect species studied suggested a lower mineral content. Previous researchers [34,35,36,38,41,46,50] found a more or less similar trend in the ash contents of different edible insect species. The energy value of any food is primarily influenced by its carbohydrate, protein and fat content, which is also true for insects. However, diet and sex are two other factors that influence insect energy value [16]. The energy content of insects can also be influenced by their developmental stages. The crude protein content ranged from 50.03 to 57.67 g/100 g in the current study, but the crude fat content of both *Laccotrephes* sp. (8.90 g/100 g) and *Ranatra* sp. (8.67 g/100 g) were significantly lower than the other three species, contributing to their lower energy value. The energy content (1389.00 to 2118.69 kJ/100 g) of all the studied species supports previous findings repeated by other researchers [34,35,36,41].

The macroelements Na, P, K, Ca, Mg and S were found in the aquatic insects studied. Edible insects, according to Kinyuru et al. [43], are a good source of the minerals that the human body requires. Phytophagous insects have lower Na content than other types of insects [51]. The Na content was found to be low (19.74 to 2.62 mg/100 g) in the current study, in contrast to previous studies on aquatic insect species in Manipur, where the Na content was reported to be 305 to 1500 mg/100 g, d.w. [36]. The obtained phosphorus content (76.34 to 153.32 mg/100 g) was consistent with the findings of edible insects in Nigeria (100.2 to 136.4 mg/100 g, d.w.) reported by Banjo et al. [46]. However, P levels in the larvae of *Cirina forda* and sun-dried edible black ants were found to be higher (215.54 mg/100 g and 158.0 to 417.0 mg/100 g, respectively) [38,42]. The highest content of phosphorus was found among the macroelements, indicating that the insects studied are a good source of P, which is a major structural component of bone in the form of a calcium phosphate salt called hydroxyapatite. The K content found in this study was quite low (22.00 to 34.60 mg/100 g) when compared to some high K-containing foods, such as almonds, avocado, whole eggs, banana, and mushrooms. Shantibala et al. [36] found a higher range of K (170 to 643 mg/100 g) in studies on aquatic insect species from Manipur. Studies on soil-dwelling Assam scarabs and edible insects in Zambia yielded comparable results to the current study, with K content ranging from 14.20 to 44.33 mg/100 g and 9.1 to 65.5 mg/100 g, respectively [34,41]. The Ca content detected in this study (32.09 to 56.15 mg/100 g) was consistent with the results obtained from aquatic insects in Manipur, where the Ca content ranged from 24.3 to 96 mg/100 g [36]. Similar results were obtained with some soil dwelling insects from Assam [41] and edible insects from Nigeria [46], where the Ca content ranged from 23.33 to 33.37 mg/100 g and 4.40 to 61.28 mg/100 g, respectively. Ca content was also similar in *C. forda* and *R. palmarum* larvae [39,42]. However, edible insects in Zambia had calcium levels as high as 166.4 mg/100 g [34]. Calcium is an important mineral for the formation of strong bones, so the studied edible insects can be a good source of Ca when combined with other Ca-rich foods. The magnesium content of the current study agrees with the Mg content of Manipur’s aquatic insects (33.6 to 99 mg/100 g) [36]. Edible insects in Zambia measured Mg up to 100 mg/100 g, which is higher than the current study [34]. Lower Mg levels (0.09 to 8.21 mg/100 g) were found in some Nigerian edible insects [46]. The S content estimated in this study ranged from 16.89 to 26.45 mg/100 g, which was found to be lower than that of a few coleopteran insects. Sulfur, an important constituent of amino acids such as methionine and cysteine, aids in the formation of di-sulfide bridges, which keep the protein intact. As a result, its inclusion in the diet is necessary.

Among the microelements studied, Fe (25.30 to 112.10 mg/100 g) was found to be abundant, comparable to soil-dwelling insects in Assam and aquatic insects in Manipur [36,41]. In comparison to red meat, insects were found to be a better source of iron [52]. When compared to Manipur aquatic insect species, zinc, magnesium, and copper were found to be in trace amounts [36]. The Zn content, on the other hand, was within the range of 2.38 to 15.86 mg/100 g found in Assam soil-dwelling scarabs [41]. Terrestrial insects, such as edible black ants, had higher Zn content in sun-dried form, ranging from 11.9 to 22.7 mg/100 g, d.w. [38]. The manganese content of aquatic insects has not previously been measured, but it is consistent with terrestrial insects such as soil-dwelling scarabs in Assam and edible insects in Zambia [34,41]. According to Bhulaidok et al. [38], the Mn content in sun-dried edible black ants was slightly high (21.0 to 32.3 mg/100 g, d.w.). Aquatic insects from Manipur had Cu content ranging from 1.1 to 13.7 mg/100 g, d.w., which was higher than the results obtained [36].

A high phenol dosage may have serious health consequences, such as decreased fertility and growth inhibition [53], and the minimum lethal oral dose of phenol for an adult is 70 mg/kg [54]. The total phenolics content in the current study ranged from 117.39 to 363.80 mg catechol equivalent/100 g, which was found to be less than the lethal oral dose and thus considered safe for consumption. Phenolic compounds are important plant constituents with redox properties responsible for antioxidant activity [55]. Flavonoids are an important antioxidant, and their presence in aquatic insects is advantageous for entomophagy. The current findings are consistent with the findings of Bhattacharyya et al. [41], who found 159 and 371 mg quercetin equivalent/100 g in *Lepidiota mansueta* and *L. albistigma*, respectively. A DPPH-scavenging activity study revealed the presence of antioxidants in the aquatic edible insects studied. A few antinutritional factors, such as tannin, phytic acid and oxalic acid, were also estimated and found to be within acceptable levels, allowing for further investigation for culinary purposes. However, Shantibala et al. [36] discovered a high tannin content in Manipur aquatic insect species. Other soil beetles, such as *L. albistigma* and *L. mansueta*, had extremely high tannin content, with 1330 mg/100 g and 324 mg/g, respectively [41]. There had been no previous reports on the phytic acid content of aquatic insects. The phytic acid content of the studied aquatic insects (11.72 to 97.30 mg/100 g) was estimated to be slightly higher than that of terrestrial insects in Nigeria such as the yam beetle and palm weevil. However, the current study’s oxalic acid content was consistent with Adesina [50]. The antinutrient content, however, was below the toxic level (0.52% or 520 mg/100 g) [36].

## 5. Conclusions

All of the aquatic insect species studied had a high nutritive value and were high in crude protein, other nutrients and energy. Significant amounts of major and trace dietary elements were also found, as well as high antioxidant activity and low antinutrients. The overall nutritional profiling of the studied species revealed their enormous potential as both human food and animal feed. Aside from the nutritional benefits offered by edible insects, the functional properties and potential applications as texturizing food ingredients and ingredients in protein-rich meat replacement products can also be investigated. Many edible insects are harvested in the wild, but more research is needed to ensure more consistent supplies through the development of economically viable methods of mass-rearing and marketing edible species.

## Figures and Tables

**Table 1 insects-13-00976-t001:** Proximate composition of the five aquatic insect species ^1^.

Insect Species	Moisture (g/100 g, f.w.) **	Carbohydrate (g/100 g, d.w.) *	Crude Protein (g/100 g, d.w.) *	Crude Fat (g/100 g, d.w.) *	Crude Fibre (g/100 g, d.w.) *	Ash Content (g/100 g, d.w.) *	Energy Content (kJ/100 g, d.w.) ***
*Diplonychus rusticus*	9.06 ± 0.38 ^a^	3.18 ± 0.19 ^abc^	57.67 ± 0.30 ^a^	27.87 ± 0.17 ^b^	2.48 ± 0.26 ^d^	4.74 ± 0.23 ^a^	2088 ± 9 ^a^
*Cybister* sp.	4.71 ± 0.26 ^c^	3.68 ± 0.18 ^a^	51.42 ± 0.31 ^d^	28.95 ± 0.20 ^a^	12.68 ± 0.23 ^a^	3.25 ± 0.23 ^b^	2118 ± 3 ^a^
*Lethocerus indicus*	3.38 ± 0.22 ^d^	2.92 ± 0.17 ^bc^	50.03 ± 0.27 ^e^	26.63 ± 0.33 ^c^	11.66 ± 0.19 ^b^	2.39 ± 0.29 ^c^	1986 ± 17 ^b^
*Laccotrephes* sp.	9.19 ± 0.23 ^a^	2.74 ± 0.23 ^c^	54.75 ± 0.34 ^c^	8.90 ± 0.22 ^d^	10.94 ± 0.40 ^b^	3.71 ± 0.18 ^b^	1389 ± 11 ^c^
*Ranatra* sp.	7.07 ± 0.11 ^b^	3.52 ± 0.23 ^ab^	56.56 ± 0.25 ^b^	8.67 ± 0.22 ^d^	9.67 ± 0.36 ^c^	3.72 ± 0.16 ^b^	1413 ± 10 ^c^

^1^ Data represent means ± standard deviations (n = 3). Values with different letters in the same column are significantly different (*p* < 0.05). * d.w.—dry weight, results obtained after drying at 103 °C except moisture content; ** f.w.—fresh weight; *** Energy value = (4 × crude protein%) + (9 × crude fat%) + (4 × carbohydrate%). Calorific values of crude protein, crude fat and carbohydrates are 4, 9 and 4, respectively.

**Table 2 insects-13-00976-t002:** Composition of macroelements of five aquatic insect species (mg/100 g, d.w.) ^1^.

Insect Species	Na *	P *	K *	Ca *	Mg *	S *
*Diplonychus rusticus*	28.62 ± 0.13 ^a^	147.16 ± 0.12 ^b^	27.78 ± 0.29 ^b^	40.13 ± 0.21 ^c^	45.20 ± 0.27 ^a^	20.90 ± 0.18 ^c^
*Cybister* sp.	22.49 ± 0.14 ^c^	153.32 ± 0.16 ^a^	34.60 ± 0.14 ^a^	32.09 ± 0.20 ^d^	38.40 ± 0.25 ^c^	16.89 ± 0.13 ^d^
*Lethocerus indicus*	26.22 ± 0.19 ^b^	120.42 ± 0.23 ^c^	23.86 ± 0.15 ^c^	48.30 ± 0.19 ^b^	38.40 ± 0.32 ^c^	22.26 ± 0.14 ^b^
*Laccotrephes* sp.	20.94 ± 0.13 ^d^	76.34 ± 0.15 ^e^	23.86 ± 0.21 ^c^	56.15 ± 0.20 ^a^	43.20 ± 0.26 ^b^	26.45 ± 0.18 ^a^
*Ranatra* sp.	19.74 ± 0.12 ^e^	114.52 ± 0.06 ^d^	22.00 ± 0.12 ^d^	32.15 ± 0.23 ^d^	33.60 ± 0.35 ^d^	20.84 ± 0.23 ^c^

^1^ Data represent means ± standard deviations (n = 3). Values with different letters in the same column are significantly different (*p* < 0.05). * d.w.—dry weight, results obtained after drying at 103 °C.

**Table 3 insects-13-00976-t003:** Composition of microelements of five aquatic insect species (mg/100 g, d.w.) ^1^.

Insect Species	Fe *	Zn *	Mn *	Cu *
*Diplonychus rusticus*	99.02 ± 0.09 ^b^	7.22 ± 0.16 ^a^	4.22 ± 0.15 ^a^	2.50 ± 0.16 ^b^
*Cybister* sp.	25.30 ± 0.14 ^e^	4.98 ± 0.21 ^d^	3.70 ± 0.20 ^a^	2.78 ± 0.20 ^b^
*Lethocerus indicus*	49.90 ± 0.15 ^d^	6.58 ± 0.19 ^b^	1.98 ± 0.22 ^c^	2.22 ± 0.14 ^b^
*Laccotrephes* sp.	90.40 ± 0.21 ^c^	5.76 ± 0.16 ^c^	2.22 ± 0.14 ^bc^	3.80 ± 0.26 ^a^
*Ranatra* sp.	112.10 ± 0.15 ^a^	6.20 ± 0.15 ^bc^	2.62 ± 0.18 ^b^	4.20 ± 0.17 ^a^

^1^ Data represent means ± standard deviations (n = 3). Values with different letters in the same column are significantly different (*p* < 0.05). * d.w.—dry weight, results obtained after drying at 103 °C.

**Table 4 insects-13-00976-t004:** Antioxidant properties of five aquatic insect species (d.w.) ^1^.

Insect Species	Total Phenolics (mg Catechol Equivalent/100 g) *	Flavonoid (mg Quercetin Equivalent/100 g) *	Antioxidant Activity (% DPPH Inhibition) *
*Diplonychus rusticus*	193.80 ± 0.29 ^c^	37.34 ± 0.27 ^c^	90.38 ± 0.47 ^b^
*Cybister* sp.	363.80 ± 0.29 ^a^	50.82 ± 0.29 ^a^	81.90 ± 0.28 ^d^
*Lethocerus indicus*	129.30 ± 0.32 ^d^	30.56 ± 0.29 ^d^	87.29 ± 0.21 ^c^
*Laccotrephes* sp.	117.39 ± 0.32 ^e^	41.69 ± 0.32 ^b^	80.82 ± 0.27 ^e^
*Ranatra* sp.	245.67 ± 0.27 ^b^	17.70 ± 0.38 ^e^	91.47 ± 0.29 ^a^

^1^ Data represent means ± standard deviations (n = 3). Values with different letters in the same column are significantly different (*p* < 0.05). * d.w.—dry weight, results obtained after drying at 103 °C.

**Table 5 insects-13-00976-t005:** Antinutritional properties of five aquatic insect species (d.w.) ^1^.

Insect Species	Tannin (mg Tannic Acid Equivalent/100 g) *	Phytic Acid (mg/100 g) *	Oxalic Acid (mg/100 g) *
*Diplonychus rusticus*	57.83 ± 0.32 ^b^	12.66 ± 0.23 ^d^	5.22 ± 0.18 ^a^
*Cybister* sp.	27.39 ± 0.26 ^d^	11.72 ± 0.26 ^e^	3.48 ± 0.30 ^c^
*Lethocerus indicus*	18.50 ± 0.31 ^e^	19.33 ± 0.32 ^c^	5.34 ± 0.24 ^a^
*Laccotrephes* sp.	36.72 ± 0.26 ^c^	97.30 ± 0.31 ^a^	4.34 ± 0.23 ^b^
*Ranatra* sp.	60.76 ± 0.27 ^a^	21.75 ± 0.29 ^b^	2.93 ± 0.17 ^c^

^1^ Data represent means ± standard deviations (n = 3). Values with different letters in the same column are significantly different (*p* < 0.05). * d.w.—dry weight, results obtained after drying at 103 °C.

## Data Availability

The data presented in this study are available on request from the corresponding author.

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
