# Peer review of "Nutritional Composition of Some Commonly Available Aquatic Edible Insects of Assam, India"

_insects, 2022, doi:10.3390/insects13110976_

Round 1

Reviewer 1 Report

The manuscript brings new original data about nutritional value of some aquatic insects in India. The idea is nice but, the paper needs to be substantially improved before it could be published.

All Latin names should be written in Italics.

SI units should be used – not kcal, but kJ, not % but g/100 g

Has the manuscript been revised by the native English speaker?

L 43: Add reference to support this information.

L 48: Nutritional value depends also on sex (Kulma et al. 2019)

L 72: Add hypothesis at the end of the Introduction.

L 81: The insects were collected in the nature, but the future of entomophagy is not eating wild insect, but insects from farms. Add the information about the potential of the tested samples to be produced via farming.

L 82: 4°C is not freezing temperature.

L 83: How long were the samples dried?

Chapter 2.2: Though you use some standard methods, say at least briefly their principles. Anthrone, Kjeldahl and Folin-Denis methods are without references. Nitrogen-to-protein conversion factor is not mentioned.

L 110: Did you analyse phenol or phenolics content?

L 120: The abbreviation is not explained.

Are the results in tables expressed per dry matter, per dried sample or per fresh weight? Please add this information.

As drying on the sun is not so standard method, I recommend recalculating all results per dry matter determined via drying at 103°C using standard method.

You do not need to explain chemical symbols of elements in the text and Table 2.

Explain the reason why you analysed antioxidant properties among nutritional composition.

Do not repeat so much the figures from the tables in the text in chapter Results.

In the Discussion, make sure that you compare your results with the results expressed on the dried sample, dry matter, or fresh sample.

Crude protein (containing also non-protein nitrogen) is not the same as protein.

You cannot compare daily requirements of protein for adult with crude protein content in insect (L 193-194). Insect can contain a reasonable amount of non-protein nitrogen.

L 206: Have you analysed dietary fibre to be able to compare your results with daily recommended intake?

L 211: The energy intake is generally influenced mainly by fat, saccharides and proteins.

L 220: Replace “is” by “are” - …minerals are essential for human body

Conclusion – add the information about the potential of farming of the analysed samples. Majority of your Conclusion is not the summary of your finding and home message from your results.

L 339: The citation is missing.

Some citated papers or chapters from book have only one page. Were all of them so short?

L 377: The citation is missing.

Author Response

The manuscript brings new original data about nutritional value of some aquatic insects in India. The idea is nice but, the paper needs to be substantially improved before it could be published.

All Latin names should be written in Italics.

Ans: All Latin names have been changed to italics in the manuscript

SI units should be used – not kcal, but kJ, not % but g/100 g

Ans: SI units have been changed from kcal to kJ and % to g/100g

Has the manuscript been revised by the native English speaker?

Ans: No

L 43: Add reference to support this information.

Ans: Reference has been added (Williams, D. and Williams, S., 2017)

L 48: Nutritional value depends also on sex (Kulma et al. 2019)

Ans: Kulma et al reference has been added

L 72: Add hypothesis at the end of the Introduction.

Ans: Hypothesis has been added: In view of that the present study was undertaken to assess the nutritive value of some selected species of edible aquatic insects based on their proximate composition, elemental composition, antioxidant properties and antinutritional properties by using standard methods of analysis which may establish aquatic insects as a potential alternate source of nutrition.

L 81: The insects were collected in the nature, but the future of entomophagy is not eating wild insect, but insects from farms. Add the information about the potential of the tested samples to be produced via farming.

Ans: Aquatic insects respond to a wide range of environmental conditions and are in-creasingly being used as water quality indicators [20]. While knowledge of terrestrial in-sect entomophagy is growing, the same cannot be said for aquatic insects [17]. There is, however, the possibility of cultivating aquatic edible species using the most basic of mate-rials, such as creating shallow artificial ponds to attract migrating adult water-beetles or raising odonates, or placing flat tiles (or pieces of 'astro-turf') in rivers and streams to col-onize by net-spinning caddisflies or blackflies [21].

L 82: 4°C is not freezing temperature.

Ans: The collected aquatic insects were later killed by freezing at 0ËšC and sorted species were washed and cleaned of dust and initially sun dried by spreading over a tray and then oven dried at 103ËšC for 4 h.

L 83: How long were the samples dried?

Ans: 4 h.

Chapter 2.2: Though you use some standard methods, say at least briefly their principles. Anthrone, Kjeldahl and Folin-Denis methods are without references. Nitrogen-to-protein conversion factor is not mentioned.

Ans: Carbohydrate content of the samples were estimated by following Anthrone Method as described by Hedge et al., [23], where the sample will be treated with 80% ethanol to remove sugars and then the saccharides is extracted with perchloric acid (HClO4). In hot acid medium saccharides is hydrolysed to glucose and dehydrated to hydroxymethyl furfural. This compound forms a green colour product with anthrone. Crude protein was estimated using Micro-Kjeldahl method [24], was calculated by multiplying the measured nitrogen content with a factor of 6.25, data were expressed as percentage (g/100g). The tannin contents were determined using Folin Denis Reagent as described by Makkar et al. [31]

L 110: Did you analyse phenol or phenolics content?

Ans: Total Phenol

L 120: The abbreviation is not explained.

Ans: Duncan's Multiple Range Test (DMRT), Statistical Package for the Social Sciences (SPSS)

Are the results in tables expressed per dry matter, per dried sample or per fresh weight? Please add this information.

Ans: Has been added in text as well as table wherever applicable

As drying on the sun is not so standard method, I recommend recalculating all results per dry matter determined via drying at 103°C using standard method.

ANS: Initially the insects were sundried to remove any water after cleaning and were later oven dried and stored

You do not need to explain chemical symbols of elements in the text and Table 2.

ANS: Corrected

Explain the reason why you analysed antioxidant properties among nutritional composition.

ANS: Antioxidants can be considered as nutrients, although they do not provide energy like other biomolecules such as carbohydrate, lipids and proteins but they are important in scavenging free radicals which are responsible damaging cells

Do not repeat so much the figures from the tables in the text in chapter Results.

Ans: Has been corrected

In the Discussion, make sure that you compare your results with the results expressed on the dried sample, dry matter, or fresh sample.

Ans: Results are been confirmed with available literature

Crude protein (containing also non-protein nitrogen) is not the same as protein.

Ans: Since, the daily requirement of protein for an adult is 0.66g/kg body weight [45] since, the edible insects of the present study has shown a high crude protein content it can be assumed that it can be added as an supplement to meet the human protein requirement.

You cannot compare daily requirements of protein for adult with crude protein content in insect (L 193-194). Insect can contain a reasonable amount of non-protein nitrogen.

Ans: Has revised it and no comparison has been done with daily requirements of protein with adults.

L 206: Have you analysed dietary fibre to be able to compare your results with daily recommended intake?

Ans: Despite the presence of the enzyme chitinase in human gastric juices, chitin is considered an indigestible fibre [47]. This enzyme, however, was discovered to be inactive. People from tropical countries with a long tradition of eating insects have a higher active chitinase response in the body [48]. Chitin removal improves the digestibility of insect protein [49].

L 211: The energy intake is generally influenced mainly by fat, saccharides and proteins.

Ans: The energy value is influenced mainly by carbohydrate, protein and fat content of any food which is also similar in case of insects. However, diet and sex are also others factors that influence the energy value in insects [16].

L 220: Replace “is” by “are” - …minerals are essential for human body

Ans: edible insects are a good source of minerals which are essential for human body.

Conclusion – add the information about the potential of farming of the analysed samples. Majority of your Conclusion is not the summary of your finding and home message from your results.

Ans: All the aquatic insect species under investigation showed high nutritive value and were rich sources of crude protein, other nutrients as well as energy.  An appreciable amount of major and trace dietary elements were also detected along with high antioxi-dant activity and low antinutrients. The overall nutritional profiling of the studied species unraveled their huge potential for further exploration as both human food and animal feed. In addition to the nutritional benefits provided by edible insects the functional prop-erties and further potential application as texturizing food ingredients as well as ingredi-ents of protein-rich meat replacing products can also be examined. In general, many edi-ble insects are harvested in the wild but research is needed to ensure more dependable supplies through development of economically feasible methods of mass-rearing and marketing of edible species.

L 339: The citation is missing.

Ans: it was not missing actually previous reference was shifted to next line

Some citated papers or chapters from book have only one page. Were all of them so short?

Ans: I have included the page numbers now. The papers were published online hence, the citation showed only one page

L 377: The citation is missing.

Ans: it was not missing actually previous reference was shifted to next line

Reviewer 2 Report

In the current manuscript,  the nutritive value of five edible aquatic insects of Assam, Indian was studied.   The article is well designed and written. The results are well presented and discussed. The manuscript needs revisions as below:

Abstract:

-write scientific names in italic form.

Introduction:

L51: add suitable reference.

-The digestibility of insect protein compared to animal and vegetable proteins should be compared.

-The importance of essential amino acids in insect protein should be mentioned.

Materials and Methods:

L82: The freezing temperature should be below zero. Check and rewrite the temperature again.

L84: percent is correct.

L85 & 90: Write the storage conditions of the samples, such as keeping them in the refrigerator, until further tests.

2.2.: write the numbers method of AOAC for analysis of each factor.

L93: What solvent was used to extract fat? Please add to the manuscript.

L103: Write the specifications of the flame photometer.

L105: Write the specifications of the atomic absorption spectroscopy.

Results:

L146: It should be stated that there was a significant difference.

L166: change " L. Indicus" to " L. indicus".

Table 5: write the letters in superscript form.

Discussion:

L173-174: In lines 83 to 84, it is mentioned that the samples were dried under the sun to below 12% moisture and the moisture content stated in the results for each sample was influenced by this initial drying, otherwise the moisture content of the insects' body is much higher than this. Therefore, the moisture content reported in previous research depends on the method of initial drying.

L197: write D. rusticus in italic form.

-The protein and fat contents of each sample should be compared with the protein and fat contents of their families that have been reported before.

- In this section, the nutritional value of the samples should be compared with the recommended daily allowance of each component or combination for each person.

L276-277: write Lepidiota mansueta and L. albistigma in italic form.

L282: write L. albistigma and L. mansueta in italic form.

L291: add a suitable reference to state the toxic level.

Author Response

In the current manuscript, the nutritive value of five edible aquatic insects of Assam, Indian was studied.   The article is well designed and written. The results are well presented and discussed. The manuscript needs revisions as below:

Abstract:

-write scientific names in italic form.

Ans: Done

Introduction:

L51: add suitable reference.

Ans: Apart from having high nutritional values, edible insects can also be considered as a fea-sible and practicable venture and means of livelihood generation, especially in rural areas as edible insects are cultivated and reared for human food and animal feed as mini-livestock [17].      

-The digestibility of insect protein compared to animal and vegetable proteins should be compared.

Ans: Exoskeletons with a high chitin content are particularly difficult to digest [12]. Indeed, we do not know whether humans can digest chitin at this time [13]. Of course, removing the exoskeleton as part of the processing is an option [14]. According to some studies, insect protein digestibility is 77%-98% without the exoskeleton [15]. The nutritional value of edi-ble insects varies widely even within the same group of species and also it depends on the metamorphic stage, habitat, their diet and sex [10, 16].

-The importance of essential amino acids in insect protein should be mentioned.

Ans: When it comes to the amino acid composition of edible insects, they have a lot of nutri-tionally valuable amino acids like phenylalanine and tyrosine. Some insects are high in lysine, tryptophan, and threonine, all of which are deficient in certain cereal proteins [5].

Materials and Methods:

L82: The freezing temperature should be below zero. Check and rewrite the temperature again.

Ans: The collected aquatic insects were later killed by freezing at 0ËšC and sorted species were washed and cleaned of dust and initially sun dried by spreading over a tray and then oven dried at 103ËšC for 4 h.

L84: percent is correct.

Ans: Corrected

L85 & 90: Write the storage conditions of the samples, such as keeping them in the refrigerator, until further tests.

Ans: The collected aquatic insects were later killed by freezing at 0ËšC and sorted species were washed and cleaned of dust and initially sun dried by spreading over a tray and then ov-en dried at 103ËšC for 4 h. The insects were then stored in airtight containers with proper labels in refrigerated condition until further tests.

2.2.: write the numbers method of AOAC for analysis of each factor.

Ans: For proximate analysis, moisture content , crude fat, crude fibre were analysed following the method as described in Association of the Official Analytical Chemists methods,2000 [22] and were expressed as percentage (g/100g).

L93: What solvent was used to extract fat? Please add to the manuscript.

Ans: Crude fat was determined using Soxhlet method using petroleum ether (40ËšC to 60ËšC) as described by AOAC method, 2000 [22]

L103: Write the specifications of the flame photometer.

Ans: It was corrected as the minerals except P was detected in AAS.

L105: Write the specifications of the atomic absorption spectroscopy.

Ans: The mineral elements sodium (Na) and potassium (K), Copper (Cu), zinc (Zn), manganese (Mn), iron (Fe), calcium (Ca), magnesium (Mg) and sulphur (S) were determined by atomic absorption spectroscopy (AAS) Atomic Absorption Spectrophotometer, (True Double Beam Optics) Model No. iCE 3500 AA Spectrometer (Wide PMT )[ 27]

L146: It should be stated that there was a significant difference.

Ans: Corrected

L166: change " L. Indicus" to " L. indicus".

Ans: Corrections done

Table 5: write the letters in superscript form.

Ans: Done

Discussion:

L173-174: In lines 83 to 84, it is mentioned that the samples were dried under the sun to below 12% moisture and the moisture content stated in the results for each sample was influenced by this initial drying, otherwise the moisture content of the insects' body is much higher than this. Therefore, the moisture content reported in previous research depends on the method of initial drying.

Ans: The collected aquatic insects were later killed by freezing at 0ËšC and sorted species were washed and cleaned of dust and initially sun dried by spreading over a tray and then ov-en dried at 103ËšC for 4 h. The insects were then stored in airtight containers with proper labels in refrigerated condition until further tests. Moisture content was determined from fresh insect sample, it was determined after drying the insect in a drying oven (Universal Hot Air Oven, Ambala Cantt, India) for 4 h at 103 °C

L197: write D. rusticus in italic form.

Ans: Correction done

-The protein and fat contents of each sample should be compared with the protein and fat contents of their families that have been reported before.

Ans: Since, there are limited study on nutritional analysis on aquatic insects hence, the study has compared it with various edible insects.

- In this section, the nutritional value of the samples should be compared with the recommended daily allowance of each component or combination for each person.

L276-277: write Lepidiota mansueta and L. albistigma in italic form.

Ans: Done

L282: write L. albistigma and L. mansueta in italic form.

Ans: Corrections done

L291: add a suitable reference to state the toxic level.

Ans: All the aquatic insect species under investigation showed high nutritive value and were rich sources of protein (50.03 to 57.67g/100g), other nutrients as well as energy.  An ap-preciable amount of major and trace dietary elements were also detected along with high antioxidant activity (80.82- 91.47% DPPH inhibition) and low antinutrients i.e below the toxic level (0.52% or 520 mg/100g) [36].

Round 2

Reviewer 1 Report

The paper still needs to be improved and its English should be corrected.

Here are my comments which have not been taken into account:

Has the manuscript been revised by the native English speaker?

Ans: No

Please, do it, or use some proofreading service, the English needs to be improved.

L 110: Did you analyse phenol or phenolics content?

Ans: Total Phenol

If you really analyse phenol (C6H5OH), please explain why you were focussed on this analyte.

As drying on the sun is not so standard method, I recommend recalculating all results per dry matter determined via drying at 103°C using standard method.

ANS: Initially the insects were sundried to remove any water after cleaning and were later oven dried and stored

Please recalculate the results in tables per dry matter determined after drying at 103°C, and add the information below the tables that results are expressed per dry matter after drying at 103°C. Also change the values in case of Energy content if you change the units from kcal to kJ.

Explain the reason why you analysed antioxidant properties among nutritional composition.

ANS: Antioxidants can be considered as nutrients, although they do not provide energy like other biomolecules such as carbohydrate, lipids and proteins but they are important in scavenging free radicals which are responsible damaging cells

Antioxidants are considered as bioactive substances, but not as nutrients. Among the nutrients there are macronutrients (lipids, saccharides, and proteins – source of energy) and micronutrients (vitamins and minerals). If you consider antioxidants as nutrients, give there the reference supporting this statement.

Crude protein (containing also non-protein nitrogen) is not the same as protein.

Ans: Since, the daily requirement of protein for an adult is 0.66g/kg body weight [45] since, the edible insects of the present study has shown a high crude protein content it can be assumed that it can be added as an supplement to meet the human protein requirement.

Correct the English: …“the edible insects of the present study have shown”….

L 206: Have you analysed dietary fibre to be able to compare your results with daily recommended intake?

Ans: Despite the presence of the enzyme chitinase in human gastric juices, chitin is considered an indigestible fibre [47]. This enzyme, however, was discovered to be inactive. People from tropical countries with a long tradition of eating insects have a higher active chitinase response in the body [48]. Chitin removal improves the digestibility of insect protein [49].

OK, and have you analysed dietary fibre via enzymatic-gravimetric method according to the AOAC method? Was the important enzymatic step included into the analytical procedure?

Author Response

Please, do it, or use some proofreading service, the English needs to be improved.

Ans: Has improved the English using quillbot application. Kindly suggest if more improvement is needed.

If you really analyse phenol (C6H5OH), please explain why you were focussed on this analyte.

 Ans: Phenolic compounds are important constituents with redox properties responsible for antioxidant activity [55]

Please recalculate the results in tables per dry matter determined after drying at 103°C, and add the information below the tables that results are expressed per dry matter after drying at 103°C. Also change the values in case of Energy content if you change the units from kcal to kJ.

Ans: Done and included in the tables

Antioxidants are considered as bioactive substances, but not as nutrients. Among the nutrients there are macronutrients (lipids, saccharides, and proteins – source of energy) and micronutrients (vitamins and minerals). If you consider antioxidants as nutrients, give there the reference supporting this statement.

Ans: Have not found such reference supporting antioxidants as nutrients. However, antioxidants are derived from these macromolecules. Kindly suggest if the manuscript topic needs change because inclusion of antioxidant in the study is very important.

Correct the English: …“the edible insects of the present study have shown”….

Ans: Have corrected the sentence

Because the daily protein requirement for an adult is 0.66g/kg body weight [45], and the edible insects in this study have a high crude protein content, it can be assumed that they can be used as a supplement to meet the human protein requirement

OK, and have you analysed dietary fibre via enzymatic-gravimetric method according to the AOAC method? Was the important enzymatic step included into the analytical procedure?

Ans: we have not analysed dietary fibre we have only analysed crude fibre 

Round 3

Reviewer 1 Report

Please, consider the following comments:

If you really analyse phenol (C6H5OH), please explain why you were focussed on this analyte.

Ans: Phenolic compounds are important constituents with redox properties responsible for antioxidant activity [55]

So please change “total phenol content” to “total phenolics content” in the whole text (lines 183, 184, 186, Table 4, 292 etc.)

Please recalculate the results in tables per dry matter determined after drying at 103°C, and add the information below the tables that results are expressed per dry matter after drying at 103°C. Also change the values in case of Energy content if you change the units from kcal to kJ.

Ans: Done and included in the tables

Yes, you add this information below the tables, but I could not find the asterisk (*) in the table.

Since the results are energy values in thousands, I recommend not expressing the values to decimal places.

Add the exact formula for calculating the energy value to chapter 2.2, or put this formula in the notes under Table 1.

Antioxidants are considered as bioactive substances, but not as nutrients. Among the nutrients there are macronutrients (lipids, saccharides, and proteins – source of energy) and micronutrients (vitamins and minerals). If you consider antioxidants as nutrients, give there the reference supporting this statement.

Ans: Have not found such reference supporting antioxidants as nutrients. However, antioxidants are derived from these macromolecules. Kindly suggest if the manuscript topic needs change because inclusion of antioxidant in the study is very important.

It is not necessary to change the title, but only make sure that you do not consider antioxidants as nutrients in the text.

OK, and have you analysed dietary fibre via enzymatic-gravimetric method according to the AOAC method? Was the important enzymatic step included into the analytical procedure?

Ans: we have not analysed dietary fibre we have only analysed crude fibre 

Please, add the description of the determination of the crude fibre to the chapter 2.2. And how could you use this crude fibre for the energy value calculation (energy content for dietary purposes should be calculated by using dietary fibre)?

Author Response

So please change “total phenol content” to “total phenolics content” in the whole text (lines 183, 184, 186, Table 4, 292 etc.)

Ans: “total phenol content” has been changed to “total phenolics content” in the whole text.

Yes, you add this information below the tables, but I could not find the asterisk (*) in the table.

Since the results are energy values in thousands, I recommend not expressing the values to decimal places.

Add the exact formula for calculating the energy value to chapter 2.2, or put this formula in the notes under Table 1.

Ans: Added the asterisk (*) in the table and removed the decimal value for energy content. Formula for calculating energy content has been put in Table 1.

It is not necessary to change the title, but only make sure that you do not consider antioxidants as nutrients in the text.

Ans: Yes, I have taken your suggestion and has not consider antioxidants as nutrients in the text.

Please, add the description of the determination of the crude fibre to the chapter 2.2. And how could you use this crude fibre for the energy value calculation (energy content for dietary purposes should be calculated by using dietary fibre)?

Ans: Description of the determination of the crude fibre has been added to the chapter 2.2. While calculating we have not considered crude fibre for energy content calculation and the correction has been made.
